# Bootstrapped Training of Score-Conditioned Generator for Offline Design of Biological Sequences

**Minsu Kim**[1]  **Federico Berto**[1]  **Sungsoo Ahn**[2]  **Jinkyoo Park**[1]

[1]Korea Advanced Institute of Science and Technology (KAIST)
[2]Pohang University of Science and Technology (POSTECH)
{min-su, fberto, jinkyoo.park}@kaist.ac.kr
{sungsoo.ahn}@postech.ac.kr

## Abstract

We study the problem of optimizing biological sequences, e.g., proteins, DNA, and RNA, to maximize a black-box score function that is only evaluated in an offline dataset. We propose a novel solution, bootstrapped training of score-conditioned generator (BOOTGEN) algorithm. Our algorithm repeats a two-stage process. In the first stage, our algorithm trains the biological sequence generator with rank-based weights to enhance the accuracy of sequence generation based on high scores. The subsequent stage involves bootstrapping, which augments the training dataset with self-generated data labeled by a proxy score function. Our key idea is to align the score-based generation with a proxy score function, which distills the knowledge of the proxy score function to the generator. After training, we aggregate samples from multiple bootstrapped generators and proxies to produce a diverse design. Extensive experiments show that our method outperforms competitive baselines on biological sequential design tasks. We provide reproducible source code: https://github.com/kaist-silab/bootgen.

## 1 Introduction

The automatic design of biological sequences, e.g., DNA, RNA, and proteins, with a specific property, e.g., high binding affinity, is a vital task within the field of biotechnology [5, 54, 41, 33]. To solve this problem, researchers have developed algorithms to optimize a biological sequence to maximize a score function [40, 9, 10, 2, 29]. Here, the main challenge is the expensive evaluation of the score function that requires experiments in a laboratory setting or clinical trials.

To resolve this issue, recent works have investigated offline model-based optimization [32, 21, 44, 52, 12, 45, MBO]. Given an offline dataset of biological sequences paired with scores, offline MBO algorithms train a proxy for the score function, e.g., a deep neural network (DNN), and maximize the proxy function without querying the true score function. Therefore, such offline MBO algorithms bypass the expense of iteratively querying the true score function whenever a new solution is proposed. However, even optimizing such a proxy function is challenging due to the vast search space over the biological sequences.

On the one hand, several works [21, 44, 52, 12] considered applying gradient-based maximization of the proxy function. However, when the proxy function is parameterized using a DNN, these methods often generate solutions where the true score is low despite the high proxy score. This is due to the fragility of DNNs against *adversarial* optimization of inputs [52, 44, 21]. Furthermore, the gradient-based methods additionally require reformulating biological sequence optimization as a continuous optimization, e.g., continuous relaxation [21, 44, 12] or mapping discrete designs to a continuous latent space [52].

On the other hand, one may consider training deep generative models to learn a distribution over high-scoring designs [32, 29]. They learn to generate solutions from scratch, which amortizes optimization over the design space.

To be specific, Kumar and Levine [32] suggests learning an inverse map from a score to a solution with a focus on generating high-scoring solutions. Next, Jain et al. [29] proposed training a generative flow network [7, GFN] as the generative distribution of high-scoring solutions.

**Contribution** In this paper, we propose a bootstrapped training of score-conditioned generator (BOOTGEN) for the offline design of biological sequences. Our key idea is to enhance the score-conditioned generator by suggesting a variation of the classical ensemble strategy of bootstrapping and aggregating. We train multiple generators using bootstrapped datasets from training and combine them with proxy models to create a reliable and diverse sampling solution.

In the bootstrapped training, we aim to align a score-conditioned generator with a proxy function by bootstrapping the training dataset of the generator. To be specific, we repeat multiple stages of (1) training the conditional generator on the training dataset with a focus on high-scoring sequences and (2) augmenting the training dataset using sequences that are sampled from the generator and labeled using the proxy function. Intuitively, our framework improves the score-to-sequence mapping (generator) to be consistent with the sequence-to-score mapping (proxy function), which is typically more accurate.

When training the score-conditioned generator, we assign high rank-based weights [46] to high-scoring sequences. Sequences that are highly ranked among the training dataset are more frequently sampled to train the generator. This leads to shifting the training distribution towards an accurate generation of high-scoring samples. Compared with the value-based weighting scheme previously proposed by Kumar and Levine [32], the rank-based weighting scheme is more robust to the change of training dataset from bootstrapping.

To further boost the performance of our algorithm, we propose two post-processing processes after the training: filtering and diversity aggregation (DA). The filtering process aims to filter samples from generators using the proxy function to gather samples with cross-agreement between the proxy and generator. On the other hand, DA collects sub-samples from multiple generators and combines them into complete samples. DA enables diverse decision-making with reduced variance in generating quality, as it collects samples from multiple bootstrapped generators.

We perform extensive experiments on six offline biological design tasks: green fluorescent protein design [54, GFP], DNA optimization for expression level on an untranslated region [41, UTR], transcription factor binding [5, TFBind8], and RNA optimization for binding to three types of transcription factors [33, RNA-Binding]. Our BOOTGEN demonstrates superior performance, surpassing the 100th percentile score and 50th percentile score of the design-bench baselines [45], a generative flow network (GFN)-based work [29] and bidirectional learning method (BDI) [12]. Furthermore, we additionally verify the superior performance of BOOTGEN in various design scenarios, particularly when given a few opportunities to propose solutions.

## 2 Related Works

### 2.1 Automatic Design of Biological Sequences

Researchers have investigated machine learning methods to automatically design biological sequences, e.g., Bayesian optimization [50, 6, 36, 38, 43], evolutionary methods [3, 8, 18, 4, 40], model-based reinforcement learning [2], and generative methods [9, 32, 20, 29, 19, 11, 14]. These methods aim to optimize biological sequence (e.g., protein, RNA, and DNA) for maximizing the target objective of binding activity and folding, which has crucial application in drug discovery and health care [29].

### 2.2 Offline Model-based Optimization

Offline model-based optimization (MBO) aims to find the design $x$ that maximizes a score function $f(x)$, using only pre-collected offline data. The most common approach is to use gradient-based optimization on differentiable proxy models trained on an offline dataset [49, 44, 52, 21, 12]. How-

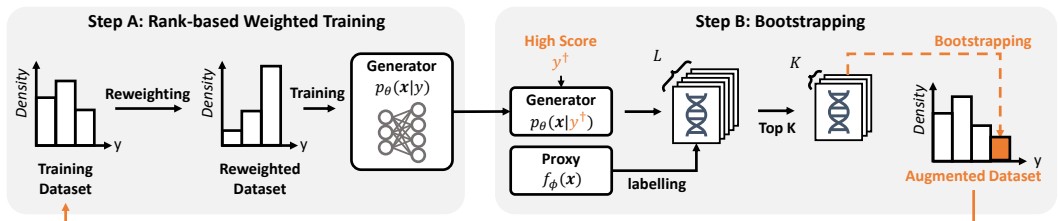

Figure 2.1: Illustration of the bootstrapped training process for learning score-conditioned generator.

ever, they can lead to poor scores due to the non-smoothness of the proxy landscape. To address this issue, Trabucco et al. [44] proposed conservative objective models (COMs), which use adversarial training to create a smooth proxy. Yu et al. [52] suggests the direct imposition of a Gaussian prior on the proxy model to create a smooth landscape and model adaptation to perform robust estimation on the specific set of candidate design space. While these methods are effective for high-dimensional continuous design tasks, their performance on discrete spaces is often inferior to classical methods, e.g., gradient ascent [45].

### 2.3 Bootstrapping

Bootstrapping means maximizing the utilization of existing resources. In a narrow sense, it refers to a statistical method where the original dataset is repeatedly sampled to create various datasets [25]. In a broader sense, it also encompasses concepts frequently used in machine learning to improve machine learning scenarios where the label is expensive, e.g., self-training and semi-supervised learning [37, 1, 22]. These methods utilize an iterative training scheme to augment the dataset with self-labeled samples with high confidence.

The bootstrapping strategy at machine learning showed great success in various domains, e.g., fully-labeled classification [51], self-supervised learning [17] and offline reinforcement learning [48]. We introduce a novel bootstrapping strategy utilizing score-conditioned generators and apply it to offline biological sequence design, addressing the challenge of working with a limited amount of poor-quality offline datasets.

### 2.4 Design by Conditional Generation

Conditional generation is a promising method with several high-impact applications, e.g., class-conditional image generation [35], language-to-image generation [39], reinforcement learning [16]. With the success of conditional generation, several studies proposed to use it for design tasks, e.g., molecule and biological sequence design.

Hottung et al. [27] proposed an instance-conditioned variational auto-encoder [31] for routing problems, which can generate near-optimal routing paths conditioned on routing instances. Igashov et al. [28] suggested a conditional diffusion model to generate 3D molecules given their fragments. Specifically, the molecular fragments are injected into the latent space of the diffusion model, and the diffusion model generates links between fragments to make the 3D molecular compound.

## 3 Bootstrapped Training of Score-Conditioned Generator (BOOTGEN)

**Problem definition** We are interested in optimizing a biological sequence $x$ to maximize a given score function $f(x)$. We consider an offline setting where, during optimization, we do not have access to the score function $f(x)$. Instead, we optimize the biological sequences using a static dataset $\mathcal{D} = \{(x_n, y_n)\}_{n=1}^N$ consisting of offline queries $y_n = f(x_n)$ to the score function. Finally, we consider evaluating a set of sequences $\{x_m\}_{m=1}^M$ as an output of offline design algorithms.

**Overview of BOOTGEN** We first provide a high-level description of our bootstrapped training of score-conditioned generator, coined BOOTGEN. Our key idea is to align the score-conditioned generation with a proxy model via bootstrapped training (i.e., we train the generator on sequences labeled using the proxy model) and aggregate the decisions over multiple generators and proxies for reliable and diverse sampling of solutions.

---

**Algorithm 1** Bootrapped Training of Score-conditioned generators

---

1: **Input:** Offline dataset $\mathcal{D} = \{\boldsymbol{x}_n, y_n\}_{n=1}^N$.
2: Update $\phi$ to minimize $\sum_{(\boldsymbol{x},y)\in\mathcal{D}}(f_\phi(\boldsymbol{x}) - y)^2$.
3: **for** $j = 1,\ldots,N_{\text{gen}}$ **do**                                              ▷ Training multiple generaters
4:     Initialize $\mathcal{D}_{\text{tr}} \leftarrow \mathcal{D}$.
5:     **for** $i = 1,\ldots,I$ **do**                                              ▷ Rank-based weighted training
6:         Update $\theta_j$ to maximize $\sum_{(\boldsymbol{x},y)\in\mathcal{D}_{\text{tr}}} w(y,\mathcal{D}_{\text{tr}}) \log p_{\theta_j}(\boldsymbol{x}|y)$.
7:         Sample $\boldsymbol{x}_\ell^* \sim p_{\theta_j}(\boldsymbol{x}|y^\dagger)$ for $\ell = 1,\ldots,L$.                 ▷ Bootstrapping
8:         Set $y_\ell^* \leftarrow f_\phi(x_\ell^*)$ for $\ell = 1,\ldots,L$.
9:         Set $\mathcal{D}_{\text{aug}}$ as top-$K$ scoring samples in $\{\boldsymbol{x}_\ell^*, y_\ell^*\}_{\ell=1}^L$.
10:         Set $\mathcal{D}_{\text{tr}} \leftarrow \mathcal{D}_{\text{tr}} \cup \mathcal{D}_{\text{aug}}$.
11:     **end for**
12: **end for**
13: **Output:** trained score-conditioned generators $p_{\theta_1}(\boldsymbol{x}|y),...,p_{\theta_{N_{\text{gen}}}}(\boldsymbol{x}|y)$.

---

Before BOOTGEN training, we pre-train proxy score function $f_\phi(\boldsymbol{x}) \approx f(\boldsymbol{x})$ only leveraging offline dataset $\mathcal{D}$. After that, our BOOTGEN first initializes a training dataset $\mathcal{D}_{\text{tr}}$ as the offline dataset $\mathcal{D}$ and then repeats the following steps:

**A.** BootGen optimizes the score-conditioned generator $p_\theta(\boldsymbol{x}|y)$ using the training dataset $\mathcal{D}_{\text{tr}}$. During training, it assigns rank-based weights to each sequence for the generator to focus on high-scoring samples.

**B.** BOOTGEN bootstraps the training dataset $\mathcal{D}_{\text{tr}}$ using samples from the generator $p_\theta(\boldsymbol{x}|y^\dagger)$ conditioned on the desired score $y^\dagger$. It uses a proxy $f_\phi(\boldsymbol{x})$ of the score function to label the new samples.

After BOOTGEN training for multiple score-conditioned generators $p_{\theta_1}(\boldsymbol{x}|y),...,p_{\theta_n}(\boldsymbol{x}|y)$, we aggregate samples from the generators with filtering of proxy score function $f_\phi$ to generate diverse and reliable samples. We provide the pseudo-code of the overall procedure in Algorithm 1 for training and Algorithm 2 for generating solutions from the trained model.

### 3.1 Rank-based Weighted Training

Here, we introduce our framework to train the score-conditioned generator. Our algorithm aims to train the score-conditioned generator with more focus on generating high-scoring designs. Such a goal is helpful for bootstrapping and evaluation of our framework, where we query the generator conditioned on a high score.

Given a training dataset $\mathcal{D}_{\text{tr}}$, our BOOTGEN minimizes the following loss function:

$$\mathcal{L}(\theta) := - \sum_{(\boldsymbol{x},y)\in\mathcal{D}_{\text{tr}}} w(y,\mathcal{D}_{\text{tr}}) \log p_\theta(\boldsymbol{x}|y), \quad w(y,\mathcal{D}_{\text{tr}}) = \frac{(k|\mathcal{D}_{\text{tr}}| + \text{rank}(y,\mathcal{D}_{\text{tr}}))^{-1}}{\sum_{(\boldsymbol{x},y)\in\mathcal{D}_{\text{tr}}}(k|\mathcal{D}_{\text{tr}}| + \text{rank}(y,\mathcal{D}_{\text{tr}}))^{-1}}.$$

where $w(y,\mathcal{D}_{\text{tr}})$ is the score-wise rank-based weight [46]. Here, $k$ is a weight-shifting factor, and $\text{rank}(y,\mathcal{D}_{\text{tr}})$ denotes the relative ranking of a score $y$ with respect to the set of scores in the dataset $\mathcal{D}_{\text{tr}}$. We note that a small weight-shifting factor $k$ assigns high weights to high-scoring samples.

For mini-batch training of the score-conditioned generator, we approximate the loss function $\mathcal{L}(\theta)$ via sampling with probability $w(y,\mathcal{D}_{\text{tr}})$ for each sample $(\boldsymbol{x},y)$.

We note that Tripp et al. [46] proposed the rank-based weighting scheme for training unconditional generators to solve online design problems. At a high level, the weighting scheme guides the generator to focus more on generating high-scoring samples. Compared to weights that are proportional to scores [32], using the rank-based weights promotes the training to be more robust against outliers, e.g., samples with abnormally high weights. To be specific, the weighting factor $w(y,\mathcal{D})$ is less affected by outliers due to its upper bound that is achieved when $\text{rank}(y,\mathcal{D}) = 1$.

---
**Algorithm 2** Aggregation Strategy for Sample Generation
---
1: **Input:** Trained score-conditioned generators $p_{\theta_1}(\boldsymbol{x}|y), ..., p_{\theta_{N_{\text{gen}}}}(\boldsymbol{x}|y)$, and trained proxy function $f_\phi(\boldsymbol{x})$.
2: Initialize $\mathcal{D}_{\text{samples}} \leftarrow \emptyset$.
3: **for** $i = 1, \ldots, N_{\text{gen}}$ **do**
4:     Sample $\boldsymbol{x}_m^* \sim p_{\theta_i}(\boldsymbol{x}|y^\dagger)$ for $m \in [M]$.
5:     Set $y_m^* \leftarrow f_\phi(x_m^*)$ for $m \in [M]$.
6:     Set $\mathcal{D}_{\text{sub-samples}}$ as Top-$K$ scoring samples in $\{\boldsymbol{x}_m^*, y_m^*\}_{m=1}^M$.         ▷ Filtering
7:     Set $\mathcal{D}_{\text{samples}} \leftarrow \mathcal{D}_{\text{samples}} \cup \mathcal{D}_{\text{sub-samples}}$         ▷ Diversity Aggregation
8: **end for**
9: **Output:** $\mathcal{D}_{\text{samples}}$.
---

## 3.2 Bootstrapping

Next, we introduce our bootstrapping strategy to augment a training dataset with high-scoring samples that are collected from the score-conditioned generator and labeled using a proxy model. Our key idea is to enlarge the dataset so that the score-conditioned generation is consistent with predictions of the proxy model, in particular for the high-scoring samples. This enables self-training by utilizing the extrapolation capabilities of the generator and allows the proxy model to transfer its knowledge to the score-conditioned generation process.

We first generate a set of samples $\boldsymbol{x}_1^*, \ldots, \boldsymbol{x}_L^*$ from the generator $p_\theta(\boldsymbol{x}|y^\dagger)$ conditioned on the desired score $y^\dagger$ [1] Then we compute the corresponding labels $y_1^*, \ldots, y_L^*$ using the proxy model, i.e., we set $y_\ell = f_\phi(\boldsymbol{x}_\ell)$ for $\ell = 1, \ldots, L$. Finally, we augment the training dataset using the set of top-$K$ samples $\mathcal{D}_{\text{aug}}$ with respect to the proxy model, i.e., we set $\mathcal{D}_{\text{tr}} \cup \mathcal{D}_{\text{aug}}$ as the new training dataset $\mathcal{D}_{\text{tr}}$.

## 3.3 Aggregation Strategy for Sample Generation

Here, we introduce additional post-hoc aggregation strategies that can be used to further boost the quality of samples from our generator. See Algorithm 2 for a detailed process.

**Filtering** We follow Kumar and Levine [32] to exploit the knowledge of the proxy function for filtering high-scoring samples from the generator. To be specific, when evaluating our model, we sample a set of candidate solutions and select the top samples with respect to the proxy function.

**Diverse aggregation** To enhance the diversity of candidate samples while maintaining reliable generating performances with low variance, we gather cross-aggregated samples from multiple score-conditioned generators. These generators are independently trained using our proposed bootstrapped training approach. Since each bootstrapped training process introduces high randomness due to varying training datasets, combining the generative spaces of multiple generators yields a more diverse space compared to a single generator.

Moreover, this process helps reduce the variance in generating quality. By creating ensemble candidate samples from multiple generators, we ensure stability and mitigate the risk of potential failure cases caused by adversarial samples. These samples may receive high scores from the proxy function but have low actual scores. This approach resembles the classical ensemble strategy known as "bagging," which aggregates noisy bootstrapped samples from decision trees to reduce variances.

# 4 Experiments

We present experimental results on six representative biological sequence design tasks to verify the effectiveness of the proposed method. We also conduct ablation studies to verify the effectiveness of each component in our method. For training, we use a single GPU of NVIDIA A100, where the training time of one generator is approximately 10 minutes.

---
[1]Following Chen et al. [12], we assume that we know the maximum score of the task.

Table 4.1: Experimental results on 100th percentile scores. The mean and standard deviation are reported for 8 independent solution generations. $\mathcal{D}$(best) indicate the maximum score of the offline dataset. The best-scored value is marked in bold.

| Method | RNA-A | RNA-B | RNA-C | TFBind8 | GFP | UTR | Avg. |
|---|---|---|---|---|---|---|---|
| $\mathcal{D}$ (best) | 0.120 | 0.122 | 0.125 | 0.439 | 0.789 | 0.593 | 0.365 |
| REINFORCE [45] | $0.462 \pm 0.080$ | $0.437 \pm 0.033$ | $0.463 \pm 0.043$ | $0.936 \pm 0.041$ | $\mathbf{0.865} \pm 0.003$ | $0.685 \pm 0.012$ | 0.643 |
| CMA-ES [24] | $0.841 \pm 0.058$ | $0.822 \pm 0.046$ | $0.803 \pm 0.039$ | $0.904 \pm 0.040$ | $0.055 \pm 0.003$ | $0.737 \pm 0.013$ | 0.694 |
| BO-qEI [50] | $0.724 \pm 0.055$ | $0.729 \pm 0.038$ | $0.707 \pm 0.034$ | $0.798 \pm 0.083$ | $0.254 \pm 0.352$ | $0.684 \pm 0.000$ | 0.649 |
| CbAS [9] | $0.541 \pm 0.042$ | $0.647 \pm 0.057$ | $0.644 \pm 0.071$ | $0.913 \pm 0.025$ | $\mathbf{0.865} \pm 0.004$ | $0.692 \pm 0.008$ | 0.717 |
| Auto. CbAS [20] | $0.524 \pm 0.055$ | $0.562 \pm 0.031$ | $0.495 \pm 0.048$ | $0.890 \pm 0.050$ | $\mathbf{0.865} \pm 0.003$ | $0.693 \pm 0.009$ | 0.672 |
| MIN [32] | $0.376 \pm 0.039$ | $0.374 \pm 0.041$ | $0.404 \pm 0.047$ | $0.892 \pm 0.060$ | $\mathbf{0.865} \pm 0.001$ | $0.691 \pm 0.011$ | 0.600 |
| Grad [45] | $0.821 \pm 0.048$ | $0.720 \pm 0.047$ | $0.688 \pm 0.035$ | $0.965 \pm 0.030$ | $0.862 \pm 0.003$ | $0.682 \pm 0.013$ | 0.792 |
| COMs [44] | $0.403 \pm 0.062$ | $0.393 \pm 0.076$ | $0.494 \pm 0.098$ | $0.945 \pm 0.033$ | $0.861 \pm 0.009$ | $0.699 \pm 0.011$ | 0.633 |
| AdaLead [42] | $0.691 \pm 0.059$ | $0.630 \pm 0.062$ | $0.605 \pm 0.055$ | $0.962 \pm 0.024$ | $0.841 \pm 0.014$ | $0.631 \pm 0.010$ | 0.727 |
| GFN-AL [29] | $0.630 \pm 0.054$ | $0.677 \pm 0.079$ | $0.623 \pm 0.045$ | $0.956 \pm 0.018$ | $0.059 \pm 0.006$ | $0.695 \pm 0.021$ | 0.607 |
| BDI [12] | $0.700 \pm 0.000$ | $0.560 \pm 0.000$ | $0.632 \pm 0.000$ | $0.973 \pm 0.000$ | $0.864 \pm 0.000$ | $0.667 \pm 0.000$ | 0.733 |
| BOOTGEN | $\mathbf{0.902} \pm 0.039$ | $\mathbf{0.931} \pm 0.055$ | $\mathbf{0.831} \pm 0.044$ | $\mathbf{0.979} \pm 0.001$ | $\mathbf{0.865} \pm 0.000$ | $\mathbf{0.865} \pm 0.000$ | $\mathbf{0.895}$ |

Table 4.2: Experimental results on 50th percentile scores. The mean and standard deviation are reported for 8 independent solution generations. $\mathcal{D}$(best) indicate the maximum score of the offline dataset. The best-scored value is marked in bold.

| Method | RNA-A | RNA-B | RNA-C | TFBind8 | GFP | UTR | Avg. |
|---|---|---|---|---|---|---|---|
| $\mathcal{D}$ (best) | 0.120 | 0.122 | 0.125 | 0.439 | 0.789 | 0.593 | 0.365 |
| REINFORCE [45] | $0.159 \pm 0.011$ | $0.162 \pm 0.007$ | $0.177 \pm 0.011$ | $0.450 \pm 0.017$ | $0.845 \pm 0.003$ | $0.575 \pm 0.018$ | 0.395 |
| CMA-ES [24] | $0.558 \pm 0.012$ | $0.531 \pm 0.010$ | $0.535 \pm 0.012$ | $0.526 \pm 0.017$ | $0.047 \pm 0.000$ | $0.497 \pm 0.009$ | 0.449 |
| BO-qEI [50] | $0.389 \pm 0.009$ | $0.397 \pm 0.015$ | $0.391 \pm 0.012$ | $0.439 \pm 0.000$ | $0.246 \pm 0.341$ | $0.571 \pm 0.000$ | 0.406 |
| CbAS [9] | $0.246 \pm 0.008$ | $0.267 \pm 0.021$ | $0.281 \pm 0.015$ | $0.467 \pm 0.008$ | $0.852 \pm 0.004$ | $0.566 \pm 0.018$ | 0.447 |
| Auto. CbAS [20] | $0.241 \pm 0.022$ | $0.237 \pm 0.009$ | $0.193 \pm 0.007$ | $0.413 \pm 0.012$ | $0.847 \pm 0.003$ | $0.563 \pm 0.019$ | 0.420 |
| MIN [32] | $0.146 \pm 0.009$ | $0.143 \pm 0.007$ | $0.174 \pm 0.007$ | $0.417 \pm 0.012$ | $0.830 \pm 0.011$ | $0.586 \pm 0.000$ | 0.383 |
| Grad [45] | $0.473 \pm 0.025$ | $0.462 \pm 0.016$ | $0.393 \pm 0.017$ | $0.513 \pm 0.007$ | $0.763 \pm 0.181$ | $0.611 \pm 0.000$ | 0.531 |
| COMs [44] | $0.172 \pm 0.026$ | $0.184 \pm 0.039$ | $0.228 \pm 0.061$ | $0.512 \pm 0.051$ | $0.737 \pm 0.262$ | $0.608 \pm 0.000$ | 0.407 |
| AdaLead [42] | $0.407 \pm 0.018$ | $0.353 \pm 0.029$ | $0.326 \pm 0.019$ | $0.485 \pm 0.013$ | $0.186 \pm 0.216$ | $0.592 \pm 0.002$ | 0.392 |
| GFN-AL [29] | $0.312 \pm 0.013$ | $0.300 \pm 0.012$ | $0.324 \pm 0.009$ | $0.538 \pm 0.045$ | $0.051 \pm 0.003$ | $0.597 \pm 0.021$ | 0.354 |
| BDI [12] | $0.411 \pm 0.000$ | $0.308 \pm 0.000$ | $0.345 \pm 0.000$ | $0.595 \pm 0.000$ | $0.837 \pm 0.010$ | $0.527 \pm 0.000$ | 0.504 |
| BOOTGEN | $\mathbf{0.707} \pm 0.005$ | $\mathbf{0.717} \pm 0.006$ | $\mathbf{0.596} \pm 0.006$ | $\mathbf{0.833} \pm 0.007$ | $\mathbf{0.853} \pm 0.017$ | $\mathbf{0.701} \pm 0.004$ | $\mathbf{0.731}$ |

## 4.1 Experimental Setting

**Tasks.** We evaluate an offline design algorithm by (1) training it on an offline dataset and (2) using it to generate 128 samples for high scores. We measure the 50th percentile and 100th percentile scores of the generated samples. All the results are measured using eight independent random seeds.

We consider six biological sequence design tasks: green fluorescent protein (GFP), DNA optimization for expression level on untranslated region (UTR), DNA optimization tasks for transcription factor binding (TFBind8), and three RNA optimization tasks for transcription factor binding (RNA-Binding-A, RNA-Binding-B, and RNA-Binding-C). The scores of the biological sequences range in $[0, 1]$. We report the statistics of the offline datasets used for each task in Table A.1. We also provide a detailed description of the tasks in Appendix A.1.

**Baselines** We compare our BOOTGEN with the following baselines: gradient ascent with respect to a proxy score model [45, Grad.], REINFORCE [49], Bayesian optimization quasi-expected-improvement [50, BO-qEI], covariance matrix adaptation evolution strategy [24, CMA-ES], conditioning by adaptative sampling [9, CbAS], autofocused CbAS [20, Auto. CbAS], model inversion network [32, MIN], where these are in the official design bench [45]. We compare with additional baselines of conservative objective models [44, COMs], generative flow network for active learning [29, GFN-AL] and bidirectional learning [12, BDI].

**Implementation** We parameterize the conditional distribution $p_\theta(x_t|\boldsymbol{x}_{1:t-1}, y)$ using a 2-layer long short-term memory [26, LSTM] network with 512 hidden dimensions. The condition $y$ is injected into the LSTM using a linear projection layer. We parameterize the proxy model using a multi-layer perceptron (MLP) with 2048 hidden dimensions and a sigmoid activation function. Our parameterization is consistent across all the tasks. We provide a detailed description of the hyperparameters in Appendix A. We also note the importance of the desired score $y^\dagger$ to condition during bootstrapping and evaluation. In this regard, we set it as the maximum score that is achievable for the given problem, i.e., we set $y^\dagger = 1$. We assume that such a value is known following [12].

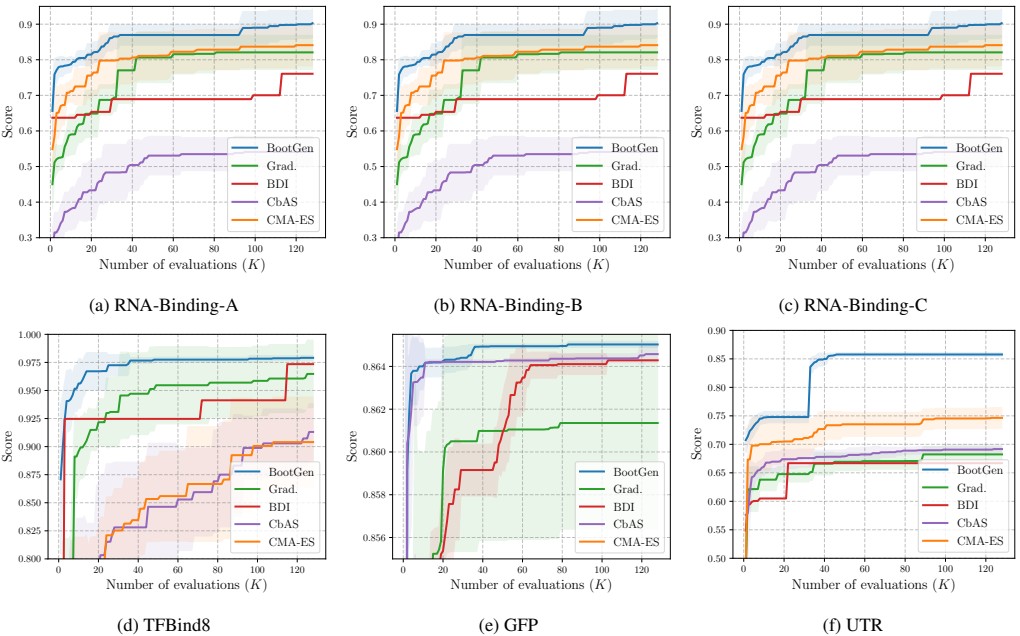

Figure 4.1: Evaluation-performance graph to compare with representative offline biological design baselines. The number of evaluations $K \in [1, 128]$ stands for the number of candidate designs to be evaluated by the Oracle score function. The average value and standard deviation error bar for 8 independent runs are reported. Our method outperforms other baselines at every task for almost all $K$.

## 4.2 Performance Evaluation

In Table 4.1 and Table 4.2, we report the performance of our BOOTGEN along with other baselines. One can observe how our BOOTGEN consistently outperforms the considered baselines across all six tasks. In particular, one can observe how our BOOTGEN achieves large gains in 50th percentile metrics. This highlights how our algorithm is able to create a reliable set of candidates.

For TFbind8, which has a relatively small search space ($4^8$), having high performances on the 100th percentile is relatively easy. Indeed, the classical method of CMA-ES and Grad. gave pretty good performances. However, for the 50th percentile score, a metric for measuring the method's reliability, BDI outperformed previous baselines by a large margin. Our method outperformed even BDI and achieved an overwhelming score.

For higher dimensional tasks of UTR, even the 50th percentile score of BOOTGEN outperforms the 100th percentile score of other baselines by a large margin. We note that our bootstrapping strategies and aggregation strategy greatly contributed to improving performances on UTR. For additional tasks of RNA, we achieved the best score for both the 50th percentile and the 100th percentile. This result verifies that our method is task-expandable.

## 4.3 Varying the Evaluation Budget

In real-world scenarios, there may be situations where only a few samples can be evaluated due to the expensive score function. For example, in an extreme scenario, for the clinical trial of a new protein drug, there may be only one or two chances to be evaluated. As we measure the 50th percentile and 100th percentile score among 128 samples following the design-bench [45] at Tables 4.1 and 4.2, we also provide a 100th percentile score report at the fewer samples from 1 sample to the 128 samples to evaluate the model's robustness on the low-budget evaluation scenarios.

To account for this, we also provide a budget-performance graph that compares the performance of our model to the baselines using different numbers of evaluations. This allows us to observe the trade-off between performance and the number of samples generated. Note that we select baselines as the Top 5 methods in terms of average percentile 100 scores reported at Table 4.1.

Table 4.3: Experimental results on 100th percentile scores (100th Per.), 50th percentile scores (50th Per.), average score (Avg. Score), diversity, and novelty, among 128 samples of UTR task. The mean and standard deviation of 8 independent runs for producing 128 samples is reported. The best-scored value is marked in bold. The lowest standard deviation is marked as the underline. The DA stands for the diverse aggregation strategy.

| Methods | 100th Per. | 50th Per. | Avg. Score | Diversity | Novelty |
|---|---|---|---|---|---|
| MIN [32] | $0.691 \pm 0.011$ | $0.587 \pm 0.012$ | $0.554 \pm 0.010$ | $28.53 \pm 0.095$ | $18.32 \pm 0.091$ |
| CMA-ES [24] | $0.746 \pm 0.018$ | $0.498 \pm 0.012$ | $0.520 \pm 0.013$ | $24.69 \pm 0.150$ | $19.95 \pm 0.925$ |
| Grad. [45] | $0.682 \pm 0.013$ | $0.513 \pm 0.007$ | $0.521 \pm 0.006$ | $25.63 \pm 0.615$ | $16.89 \pm 0.426$ |
| GFN-AL [29] | $0.700 \pm 0.015$ | $0.602 \pm 0.014$ | $0.580 \pm 0.014$ | $30.89 \pm 1.220$ | $20.25 \pm 2.272$ |
| BootGen w/o DA. | $0.729 \pm 0.074$ | $0.672 \pm 0.082$ | $0.652 \pm 0.081$ | $17.83 \pm 5.378$ | $20.49 \pm 1.904$ |
| BootGen w/ DA. (*ours*) | $\mathbf{0.858} \pm \underline{0.003}$ | $\mathbf{0.701} \pm \underline{0.004}$ | $\mathbf{0.698} \pm \underline{0.001}$ | $\mathbf{31.57} \pm \underline{0.073}$ | $\mathbf{21.40} \pm \underline{0.057}$ |

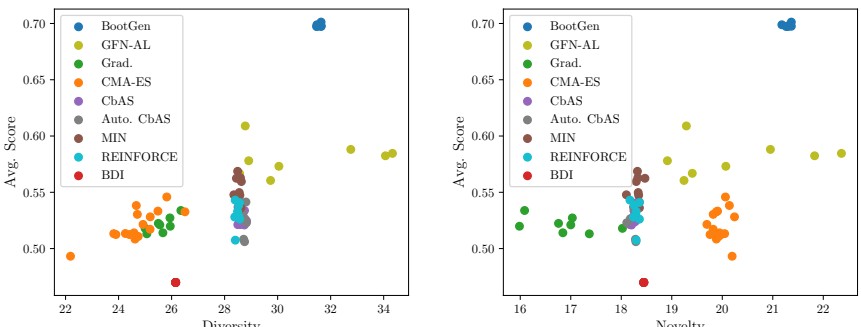

Figure 4.2: Multi-objectivity comparison of diversity and novelty on the average score for the UTR task. Each datapoint for 8 independent runs is depicted.

As shown in Fig. 4.1, our method outperforms every baseline for almost every evaluation budget. For the UTR task, our performance on a single evaluation budget gives a better score than the other baselines' scores when they have a budget of 128 evaluations. For RNA tasks, our method with an approximate budget of 30 achieves superior performance compared to other methods with a budget of 128. These results show that our method is the most reliable as its performance is most robust when the evaluation budget is limited.

## 4.4 Average Score with Diversity

For biological sequence design, measuring the diversity and novelty of the generated sequence is also crucial [29]. Following the evaluation metric of [29] we compare the performance of models in terms of diversity and novelty.

Here is measurement of diversity for sampled design dataset $\mathcal{D} = \{\boldsymbol{x}_1, ..., \boldsymbol{x}_M\}$ from generator which is average of *Levenshtein distance* [23], denoted by $d(\boldsymbol{x}_i, \boldsymbol{x}_j)$, between arbitrary two biological sequences $\boldsymbol{x}_i, \boldsymbol{x}_j$ from the generated design candidates $\mathcal{D}$:

$$\text{Diversity}(\mathcal{D}) := \frac{1}{|\mathcal{D}|(|\mathcal{D}| - 1)} \sum_{\boldsymbol{x} \in \mathcal{D}} \sum_{\boldsymbol{s} \in \mathcal{D} \setminus \{\boldsymbol{x}\}} d(\boldsymbol{x}, \boldsymbol{s}).$$

Next, we measure the minimum distance from the offline dataset $\mathcal{D}_{\text{offline}}$ which measures the novelty of generated design candidates $\mathcal{D}$ as:

$$\text{Novelty}(\mathcal{D}, \mathcal{D}_{\text{offline}}) = \frac{1}{|\mathcal{D}|} \sum_{\boldsymbol{x} \in \mathcal{D}} \min_{\boldsymbol{s} \in \mathcal{D}_{\text{offline}}} d(\boldsymbol{x}, \boldsymbol{s}).$$

Our method surpasses all baselines, including GFN-AL [29], in the UTR task, as evidenced by the Pareto frontier depicted in Table 4.3 and Fig. 4.2. Given the highly dimensional nature of the UTR task and its expansive search space, the discovery of novel and diverse candidates appears to be directly related to their average score. This implies that extensive exploration of the high-dimensional space is crucial for improving scores in the UTR task.

Table 4.4: Ablation study for BOOTGEN. The average score among 128 samples is reported. We make 8 independent runs to produce 128 samples where the mean and the standard deviation are reported. For every method, an aggregation strategy is applied by default. The best-scored value is marked in bold. The lowest standard deviation is underlined. The RR stands for rank-based reweighting, the B stands for bootstrapping, and the F stands for filtering.

| Components | RNA-A | RNA-B | RNA-C | TFbind8 | UTR | GFP |
|---|---|---|---|---|---|---|
| $\emptyset$ | $0.388 \pm 0.007$ | $0.350 \pm 0.008$ | $0.394 \pm 0.010$ | $0.579 \pm 0.010$ | $0.549 \pm 0.009$ | $0.457 \pm 0.044$ |
| $\{RR\}$ | $0.483 \pm 0.006$ | $0.468 \pm 0.008$ | $0.441 \pm 0.010$ | $0.662 \pm 0.009$ | $0.586 \pm 0.008$ | $0.281 \pm 0.031$ |
| $\{RR, B\}$ | $0.408 \pm 0.009$ | $0.379 \pm 0.009$ | $0.417 \pm 0.006$ | $0.666 \pm 0.009$ | $0.689 \pm 0.003$ | $0.470 \pm 0.034$ |
| $\{RR, F\}$ | $0.576 \pm 0.005$ | $0.586 \pm 0.004$ | $0.536 \pm 0.007$ | $0.833 \pm 0.004$ | $0.621 \pm 0.003$ | $0.783 \pm 0.011$ |
| $\{RR, F, B\}$ | $\mathbf{0.607} \pm 0.009$ | $\mathbf{0.612} \pm 0.005$ | $\mathbf{0.554} \pm 0.007$ | $\mathbf{0.840} \pm 0.004$ | $\mathbf{0.698} \pm 0.001$ | $\mathbf{0.804} \pm 0.002$ |

It is worth noting that GFN-AL, which is specifically designed to generate diverse, high-quality samples through an explorative policy, secures a second place for diversity. Although GFN-AL occasionally exhibits better diversity than our method and achieves a second-place average score, it consistently delivers poor average scores in the GFP and RNA tasks Table 4.2. This drawback can be attributed to its high explorative policy, which necessitates focused exploration in narrow regions. In contrast, BOOTGEN consistently produces reliable scores across all tasks Table 4.2. For a comprehensive comparison with GFN-AL, please refer to the additional experiments presented in Appendix C.

**Diverse aggregation strategy** Our diverse aggregation (DA) strategy significantly enhances diversity, novelty, and score variance, as demonstrated in Table 4.3. This is especially beneficial for the UTR task, which necessitates extensive exploration of a vast solution space, posing a substantial risk to the bootstrapped training process. In this context, certain bootstrapped generators may yield exceedingly high scores, while others may produce low scores due to random exploration scenarios. By employing DA, we combine multiple generators to generate candidate samples, thereby greatly stabilizing the quality of the bootstrapped generator.

## 4.5 Ablation study

The effectiveness of our components, namely rank-based reweighting (RR), bootstrapping (B), and filtering (F), in improving performance is evident in Table 4.4. Across all tasks, these components consistently contribute to performance enhancements. The bootstrapping process is particularly more beneficial for high-dimensional tasks like UTR and GFP. This correlation is intuitive since high-dimensional tasks require a larger amount of data for effective exploration. The bootstrapped training dataset augmentation facilitates this search process by leveraging proxy knowledge. Additionally, the filtering technique proves to be powerful in improving scores. As we observed from the diverse aggregation and filtering, the ensemble strategy greatly enhances score-conditioned generators.

## 5 Future Works

**Enhancing proxy robustness** While our bootstrapping method shows promise for offline bio-sequential design tasks, it has inherent technical limitations. The assumption that the generator produces superior data to the training dataset may backfire if the generator samples have poor quality designs and the proxy used is inaccurate. While the current aggregation strategy effectively manages this risk, we can address this limitation by utilizing robust learning methods of proxies such as conservative proxies modeling [44], robust model adaptation techniques [52], parallel mentoring proxies [13], and importance-aware co-teaching of proxies [53] for further improvement.

**Enhancing architecture of BOOTGEN** Our approach primarily employs a straightforward architectural framework, with a primary emphasis on validating its algorithmic structures in the context of offline biological sequence design. To enhance the practical utility of our method, it will be advantageous to incorporate established and robust architectural paradigms, exemplified in works such as [11] and [14], into the framework of our method. One promising avenue for achieving this integration is the incorporation of pre-trained protein language models (pLMs) [34, 15], akin to those expounded upon in [14].

# 6   Conclusion

This study introduces a novel approach to stabilize and enhance score-conditioned generators for offline biological sequence design, incorporating the classical concepts of bootstrapping and aggregation. Our novel method, named BOOTGEN, consistently outperformed all baselines across six offline biological sequence design tasks, encompassing RNA, DNA, and protein optimization. Our strategy of bootstrapping and aggregation yielded remarkable improvements in achieving high scores, generating diverse samples, and minimizing performance variance.

## Acknowledgements

We thank all the valuable comments and suggestions from anonymous reviewers who helped us improve and refine our paper. This work was supported by the Institute of Information & communications Technology Planning & Evaluation (IITP) grant funded by the Korean government(MSIT)(2022-0-01032, Development of Collective Collaboration Intelligence Framework for Internet of Autonomous Things).

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

# A    Additional Experimental Settings

Table A.1: Details of the offline datasets. We let $|\mathcal{X}|$ and $|\mathcal{D}|$ denote the sizes of the search space and the offline dataset, respectively.

|  | Seq. Length | Vocab size | $|\mathcal{X}|$ | $|\mathcal{D}|$ |
|---|---|---|---|---|
| GFP | 20 | 237 | $20^{237}$ | 5,000 |
| UTR | 50 | 4 | $50^4$ | 140,000 |
| TFBind8 | 8 | 8 | $4^8$ | 32,898 |
| RNA-Binding | 14 | 4 | $4^{14}$ | 5,000 |

## A.1    Datasets

- **GFP** [41] is a task to optimize a protein sequence of length 237 consisting of one of 20 amino acids, i.e., the search space is $20^{237}$. Its objective is to find a protein with high fluorescence. Following Trabucco et al. [45], we prepare the offline dataset using 5000 samples with 50 to 60 percentile scores in the original data.

- **UTR** [41] is a task to optimize a DNA sequence of length 50 consisting of one of four nucleobases: adenine (A), guanine (G), cytosine (C), thymine (T). Its objective is to maximize the expression level of the corresponding 5'UTR region. For the construction of the offline dataset $\mathcal{D}$, following Trabucco et al. [45], we provide samples with scores under the 50th percentile data of $140,000$ examples.

- **TFBind8** [5] is a task that optimizes DNA similar to the UTR. The objective is to find a length 8 sequence to maximize the binding activity with human transcription factors. For the offline dataset $\mathcal{D}$, we provide under 50th percentile data of 32,898 examples following Trabucco et al. [45].

- **RNA-Binding** [33] is a task that optimizes RNA, a sequence that contains four vocab words of nucleobases: adenine (A), uracil (U), cytosine (C), and guanine (G). The objective is to find a length 14 sequence to maximize the binding activity with the target transcription factor. We present three target transcriptions of RNA termed RNA-Binding-A (for L14 RNA1), RNA-Binding-B (for L14 RNA2), and RNA-Binding-C (for L14 RNA3). We provide under 0.12 scored data for the offline dataset $\mathcal{D}$ among randomly generated 5,000 sequences using open-source code [2].

## A.2    Implementation of Baselines

This section provides a detailed implementation of baselines of offline biological sequence design.

**Baselines from Design Bench [45].**    Most baselines are from the offline model-based optimization (MBO) benchmark called design-bench [45]. The design bench contains biological sequence tasks of the GFP, UTR, and TFbind8, where it contains baselines of REINFORCE, CMA-ES [24], BO-qEI [50], CbAS [9], Auto. Cbas [20], MIN [32], gradient ascent (Grad.), and COMS [44]. We reproduce them by following the official source code [3]. For the RNA tasks, we follow hyperparameters of TFBind8 as the number of vocab are same as 4, and the sequence length is similar where the TFBind8 has length 8 and RNA tasks have length 14 as our method follows the same.

**BDI [12].**    For BDI, we follow hyperparameter setting at the paper [12] and implementation at the opensource code [4]. For RNA tasks, we follow the hyperparameter for TFBind8 tasks, as our method follows the same.

**GFN-AL [29].**    For GFN-AL we follow hyperparameters setting at the paper [29] and implementation on open-source code [5]. Because they only reported the TFbind8 and the GFP tasks, we use the hyperparameter of the GFP for the UTR tasks hyperparameter of the TFBind8 for RNA tasks, as our method follows the same.

---

[2] https://github.com/samsinai/FLEXS
[3] https://github.com/brandontrabucco/design-baselines
[4] https://github.com/GGchen1997/BDI
[5] https://github.com/MJ10/BioSeq-GFN-AL

## A.3 Hyperparameters

Table A.2: Hyperparameters. $I$ denotes the number of bootstrapping iterations after pretraining, $I'$ denotes the number of pretaining iterations, $M$ represents the number of samples used in inference, $K$ stands for the number of filtered samples among $M$ candidates, and $N_{\text{gen}}$ refers to the number of aggregated generators in the experiments.

| $I$ | $I'$ | $M$ | $K$ | $N_{\text{gen}}$ |
|---|---|---|---|---|
| 2,500 | 12,500 | 1,280 | 128 | 8 |

**Training.** We give consistency hyperparameters for all tasks except the learning rate. We set the generator's learning rate to $10^{-5}$ for short-length tasks (lengths 8 and 14) of TFBind and RNA tasks and $5 \times 10^{-5}$ for longer-length tasks (lengths 50 and 237) of UTR and GFP. We trained the generator with 12,500 steps before bootstrapping. Bootstrapping is applied with $2,500$ in additional steps. The batch size of training is 256. We set the weighting parameter $k = 10^{-2}$. Note that we early-stopped the generator iteration of GFP with the $3,000$ step based on monitoring the calibration model of Appendix B. For bootstrapping, the generator samples 2 candidates every 5 steps. For Top-K sampling at the bootstrapping, we sample with $L = 1,000$ and select the Top 2 samples to augment the training dataset.

**Testing.** For filtering, we generated $M = 1,280$ candidate samples and collected the Top-K samples where $K = 128$ based on the proxy score. For diverse aggregation, we collect $K = 16$ samples from 8 generators, making a total of 128 samples.

**Proxy model.** For the proxy model, we applied a weight regularization of $10^{-4}$, set the learning rate to $10^{-4}$, and used a dropout rate of 0.1. We used early stopping with a tolerance of 5 and a train/validate ratio of 9:1 following Jain et al. [29]. We used the Adam optimizer [30] for the training generator, proxy, and calibration model.

# B Calibration Model

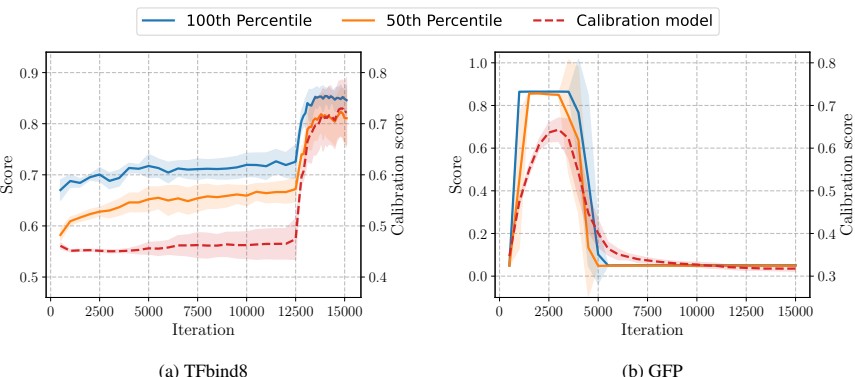

(a) TFbind8         (b) GFP

Figure B.1: Calibration model's tendency.

Tuning the hyperparameters of offline design algorithms is challenging due to the lack of access to the true score function. Therefore, existing works have proposed various strategies to circumvent this issue, e.g., choosing a hyperparameter that is transferrable between different tasks [45] or tuning the hyperparameter based on training statistics [52].

In this work, we leverage the calibration function. Inspired by Wang et al. [47], we train the calibration function on the offline dataset to approximate the true score function similar to the proxy function. Then we use the calibration function to select a score-conditioned model that achieves higher performance with respect to the calibration function. We also choose the number of training steps and early stopping points using the same criterion.

As shown in Fig. B.1, the calibration model accurately predicts early stopping points as the GFP task is unstable and has a narrow high score region which gives a high chance to be overfitted into the low-scored region (Table 4.1 shows that score of GFP is highly polarized). By using the calibration function, we can simply choose an early stopping point for GFP. Note we simply leverage the proxy model as a calibration model with an exact sample training scheme and hyperparameters.

# C  Diversity and Novelty Comparison with GFN-AL [29]

Table C.1: Experimental results on 100th percentile scores (100th Per.), 50th percentile scores (50th Per.), average score (Avg. Score), diversity, and novelty, among 128 samples of low dimensional tasks comparing with GFN-AL. The mean and standard deviation of 8 independent runs for producing 128 samples is reported. The best-scored value is marked in bold.

| | Methods | 100th Per. | 50th Per. | Avg. Score | Diversity | Novelty |
|---|---|---|---|---|---|---|
| RNA-A | GFN-AL | $0.630 \pm 0.054$ | $0.312 \pm 0.013$ | $0.320 \pm 0.010$ | $8.858 \pm 0.045$ | $4.269 \pm 0.130$ |
| | BootGen | $\mathbf{0.898} \pm 0.039$ | $\mathbf{0.694} \pm 0.009$ | $\mathbf{0.699} \pm 0.008$ | $5.694 \pm 0.008$ | $\mathbf{7.509} \pm 0.049$ |
| | BootGen$^\dagger$ | $0.750 \pm 0.041$ | $0.382 \pm 0.014$ | $0.399 \pm 0.008$ | $\mathbf{8.917} \pm 0.078$ | $4.957 \pm 0.052$ |
| RNA-B | GFN-AL | $0.677 \pm 0.080$ | $0.300 \pm 0.012$ | $0.311 \pm 0.011$ | $8.846 \pm 0.050$ | $4.342 \pm 0.128$ |
| | BootGen | $\mathbf{0.886} \pm 0.028$ | $\mathbf{0.689} \pm 0.007$ | $\mathbf{0.693} \pm 0.007$ | $5.192 \pm 0.073$ | $\mathbf{7.981} \pm 0.036$ |
| | BootGen$^\dagger$ | $0.686 \pm 0.052$ | $0.355 \pm 0.018$ | $0.371 \pm 0.017$ | $\mathbf{8.929} \pm 0.121$ | $4.967 \pm 0.132$ |
| RNA-C | GFN-AL | $0.623 \pm 0.045$ | $0.324 \pm 0.010$ | $0.333 \pm 0.010$ | $8.831 \pm 0.046$ | $4.151 \pm 0.088$ |
| | BootGen | $\mathbf{0.837} \pm 0.045$ | $\mathbf{0.598} \pm 0.006$ | $\mathbf{0.606} \pm 0.006$ | $4.451 \pm 0.071$ | $\mathbf{7.243} \pm 0.051$ |
| | BootGen$^\dagger$ | $0.651 \pm 0.056$ | $0.370 \pm 0.011$ | $0.376 \pm 0.013$ | $\mathbf{8.913} \pm 0.087$ | $4.597 \pm 0.073$ |
| TFBind8 | GFN-AL | $0.951 \pm 0.026$ | $0.537 \pm 0.055$ | $0.575 \pm 0.037$ | $5.001 \pm 0.178$ | $0.778 \pm 0.143$ |
| | BootGen | $\mathbf{0.977} \pm 0.004$ | $\mathbf{0.848} \pm 0.010$ | $\mathbf{0.839} \pm 0.009$ | $3.118 \pm 0.045$ | $\mathbf{1.802} \pm 0.025$ |
| | BootGen$^\dagger$ | $0.970 \pm 0.018$ | $0.613 \pm 0.017$ | $0.627 \pm 0.014$ | $\mathbf{5.048} \pm 0.039$ | $0.965 \pm 0.033$ |

Table C.2: Experimental results on 100th percentile scores (100th Per.), 50th percentile scores (50th Per.), average score (Avg. Score), diversity, and novelty, among 128 samples of 6 six biological sequential tasks comparing with GFN-AL. The mean and standard deviation of 8 independent runs for producing 128 samples is reported. The best-scored value is marked in bold. The 'Random' stands for uniform random generator.

| | Methods | 100th Per. | 50th Per. | Avg. Score | Diversity | Novelty |
|---|---|---|---|---|---|---|
| GFP | Random | $0.053 \pm 0.000$ | $0.051 \pm 0.000$ | $0.051 \pm 0.000$ | $\mathbf{219.840} \pm 0.207$ | $\mathbf{216.960} \pm 0.330$ |
| | GFN-AL | $0.057 \pm 0.001$ | $0.051 \pm 0.004$ | $0.052 \pm 0.004$ | $130.113 \pm 41.202$ | $208.610 \pm 46.271$ |
| | BootGen | $\mathbf{0.865} \pm 0.000$ | $\mathbf{0.854} \pm 0.002$ | $\mathbf{0.813} \pm 0.011$ | $7.969 \pm 0.460$ | $2.801 \pm 0.163$ |

Building upon the § 4.4 findings of the UTR, we present further multi-objective experimental results for the remaining 5 tasks, comparing them closely with the GFN-AL [29] approach. The GFN-AL model aims to achieve extensive exploration by prioritizing sample diversity and novelty, leading to the generation of diverse, high-quality biological sequences. Nevertheless, the diversity measure occasionally introduces a trade-off between sample scores, particularly when certain tasks exhibit a narrow score landscape, resulting in only a limited number of samples with high scores.

We conducted a detailed analysis to shed light on the relationship between score metrics (average, 100th percentile, 50th percentile) and diversity metrics (diversity and novelty). The results, presented in Table C.2, demonstrate that our proposed method, BootGen, outperforms GFN-AL in terms of score performance. However, it is noteworthy that GFN-AL exhibits high diversity, particularly in the case of GFP. On the contrary, GFN-AL generates extremely low scores for GFP, almost comparable to those produced by a uniform random generator. This observation indicates that the GFP task possesses a narrow score landscape, making it relatively easy to generate diverse yet low-scoring samples.

For the TFbind8 and RNA tasks, GFN-AL achieves high diversity but a low novelty. This suggests that GFN-AL struggles to discover samples beyond the scope of the offline training dataset, resulting in less novel but diverse samples with low scores. In contrast, BootGen successfully identifies high-scoring and novel samples. Consequently, in this scenario, we consider high diversity coupled with low novelty and score to be somewhat meaningless, as such results can also be achieved by a random generator.

To substantiate our claim regarding diversity, we present experimental results of an enhanced diversity version of BootGen. In order to achieve increased diversity, BootGen makes certain sacrifices in terms of score performance. One approach we employ is interpolation with a uniform random sequence generator. Specifically, we combine our generator with the uniform random gen-

erator to generate random samples in a portion of the sequence (we make $3/4$ samples from the random generator and $1/4$ from BOOTGEN). Additionally, we can filter out low-diversity sequences without requiring score evaluation, thereby generating a more diverse set of samples by referring code of GFN-AL [29]. To this end, we introduce the diversity-improved version of our method, denoted as BOOTGEN$^{\dagger}$. It is important to note that BOOTGEN$^{\dagger}$ sacrifices score performance, as diversity and score are inherent trade-offs, and it focuses primarily on diversity to provide a more direct comparison with GFN-AL by manually adjusting diversity.

As shown in Table C.2, BOOTGEN$^{\dagger}$ exhibits similar diversity levels in RNA-A, RNA-B, RNA-C, and TFBind8, while achieving higher score metrics and novelty. We attribute these results to GFN's underfitting issue, as it fails to adequately fit within the high-scoring region of the score landscape, particularly for the high-dimensional tasks of UTR and GFP.

# D   Rank-based weighting vs. Value-based weighting

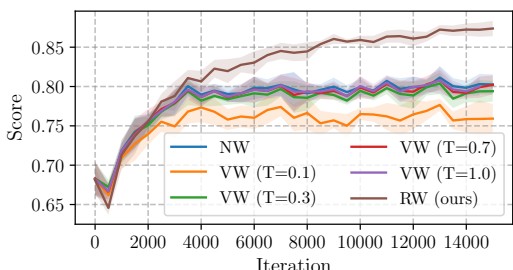

Figure D.1: Comparison of rank-based weighting (RW) and value-based weighting (VW) methods. The NW represents the case where no weighting is applied to the training distribution. In the VW case, we explored different weighting temperatures, $T \in \{0.1, 0.3, 0.7, 1.0\}$. The 50th percentile scores of TFBind8 are reported, and the results include a bootstrapping procedure applied from iteration 12,500 to 15,000.

We verify the contribution of the rank-based weighting (RW) scheme compared to ours with no weighting (NW) and the existing value-based weighting (VW) proposed by Kumar and Levine [32]. To implement VW, we set the sample-wise weight proportional to $\exp(|y - y^*|/T)$, where $y^*$ is the maximum score in the training dataset and $T \in \{0.1, 0.3, 0.7, 1.0\}$ is a hyperparameter. As shown in Fig. D.1, the results indicate that RW outperforms both NW and VW. This validates our design choice for BOOTGEN.

