# OpenReview forum: "Bootstrapped Training of Score-Conditioned Generator for Offline Design of Biological Sequences"
_NeurIPS.cc/2023/Conference — NeurIPS 2023 poster_

### Official Review · Reviewer_M7pC · 2023-06-15

**Soundness:** 2 fair
**Presentation:** 3 good
**Contribution:** 3 good
**Rating:** 7
**Confidence:** 3

**Summary:**

This paper introduces a novel algorithm, the Bootstrapped Training of Score-Conditioned Generator (BOOTGEN), to optimize the design of biological sequences. BOOTGEN overcomes the challenge of high-cost evaluations and vast search space by training a score-based generator using rank-based weights and a bootstrapping process. This generator is then augmented with self-generated data labeled by a proxy score function. The process results in diverse and accurate biological sequence designs. The efficacy of BOOTGEN is demonstrated via extensive experiments across six offline biological design tasks.

**Strengths:**

The paper is clear and well-presented. The authors provide a comprehensive introduction and explanation of the BOOTGEN algorithm, making it easily understandable even for those who may not be specialists in this specific field. The experimental results are also clearly laid out and explained, adding to the overall lucidity of the paper.

Furthermore, the extensive experiments and superior results over existing methods reinforce the potential value of this work to both academic researchers and industry practitioners.

**Weaknesses:**

1. The authors have overlooked a crucial piece of literature, "Deep Extrapolation for Attribute-Enhanced Generation" (https://arxiv.org/abs/2107.02968), presented at NeurIPS 2021. This paper also explores protein sequence design and employs a generator/scorer framework. The authors need to acknowledge this work and conduct comparative analysis with it.

2. The first word of the title "Automatic design of biological sequences" should be capitalized. This correction should also be made in Section 4.2, where the title reads "Varying the evaluation budget."

3. Claiming the ranking-based weighting as a significant contribution over the value-based weighting is not entirely novel and may not warrant its inclusion as a key contribution.

4. The paper's differentiation from "Conditioning by adaptive sampling for robust design," which also uses a generative model and a proxy for labeling, is unclear.

**Questions:**

See Weaknesses.

**Limitations:**

Limitations discussed.

---

> ### Author Rebuttal · Authors · 2023-08-06
>
> Thanks for providing a valuable review.
>
> **W1: About the paper "Deep Extrapolation for Attribute-Enhance Generation"**
>
>
> Thank you for pointing out the relevant literature. GENhance [1] and our method share the common goal of extrapolating from offline datasets using generative models. BootGen is a high-level algorithm that leverages existing (conditional) generative models. In contrast, GENhance is a novel generative model with significant neural architecture and learning loss. Therefore those works are orthogonal. The combination of GENhance and BootGen seems complementary, presenting a promising direction for future work. We will dedicate a discussion section further to analyze this integration's potential benefits and implications.
>
> ---
>
> **W2: About typo**
>
> Thank you for pointing out the typo; we will revise it.
>
> ---
>
> **W3: Contribution claims of rank-based weighting**
>
> Thank you for your understanding. While we acknowledge the differences from applying rank-based weighting to a conditional generator, we agree that this aspect alone may not be the essential contribution. Our principal gifts lie in the novel combination of techniques and the high-level algorithmic structure, which facilitates stable extrapolation for offline biological sequence optimization. We highly value your feedback and will emphasize these core contributions in our paper.
>
> ---
>
> **W4: Unclear differentiation from "Conditioning by adaptive sampling for robust design,**
>
> Our algorithm addresses the challenge of offline black-box optimization through a combination of reweighting processes and extrapolation of queries for score-conditioned models. The objective is to generate novel samples beyond the scope of the offline dataset without relying on an oracle function. To achieve this, we employ an aggregation method to ensure the stability of the process.
>
> The CbAS [2] method is a representative black-box optimization technique that excels in generating candidate samples to maximize the oracle function in an online setting. However, when applied to offline scenarios, it falls short in performance. Unlike CbAS, which relies on continuous updates from online Oracle queries, our approach leverages reweighting and extrapolation processes to excel in offline settings. Tables 1 and 2 in the main text demonstrate our superior performance across all tasks compared to CbAS.
>
>
> ---
>
> ### References
>
> [1] Chan, Alvin, et al. "Deep extrapolation for attribute-enhanced generation." Advances in Neural Information Processing Systems 34 (2021): 14084-14096.
>
> [2] Brookes, David, Hahnbeom Park, and Jennifer Listgarten. "Conditioning by adaptive sampling for robust design." International conference on machine learning. PMLR, 2019.

---

> > ### Comment · Reviewer_M7pC · 2023-08-10
> > **Feedback**
> >
> > I still think that CbAS is similar to your work. CbAS can also be applied in this offline scenario. It also leverages a proxy to learn a better generator to extrapolate.

---

> > > ### Author Response · Authors · 2023-08-11
> > >
> > > Thank you for fostering an insightful discussion.
> > >
> > > As you aptly pointed out, there is a common aim between CbAS and BootGen. Both methods share core principles, including proxy utilization, online training through proxy and generator interplay, and the incorporation of weighted Maximum Likelihood Estimation (MLE) training.
> > >
> > > To elucidate the distinctions between CbAS and BootGen:
> > >
> > > |  | CbAS | BootGen |
> > > | -------- | -------- | -------- |
> > > | Underlying Distribution     | $q(x)$     | $p(xㅣy)$     |
> > > | Generative Framework   | Weighted VAE     | Conditional Auto-regressive Model     |
> > > | Ensemble  | Proxy Model (ensemble for y)    | Generative Model (ensemble for x)   |
> > > | Weighting Approach | Conditional probability to enforce desired value set S given x     | Assessment through dataset ranking   |
> > >
> > >
> > >
> > > CbAS trains estimate conditional distribution $p(x|S)$ by introducing variational distribution $q(x;\phi)$, which is trained to minimize $D_{KL}(p(x|S)||q(x)) = E_{p(x)}[P(S|x)logq(x)]$ where $S$ stands for a set of desired property value. This can be seen as weighted MLE where  $P(S|x)$ stands for weight; i.e., high probability for desired property value gives high weight.  The weight $P(S|x)$ is estimated using the Oracle score function (or proxy function can be used in the offline setting, as you mentioned).  After training $q(x;\phi)$, desired valued designs are sampled from the variational distribution: $x \sim q(x;\phi)$.
> > >
> > > In contrast, BootGen directly trains **conditional** distribution $p(x|y;\phi)$ using weighted MLE (rank-based weight) and periodically augment offline dataset using a training generator and proxy model. The design is sampled from conditional distribution by querying desired value in the inference time: $x \sim p(x|y=y^*;\phi)$. The ensemble process is done for aggregating multiple generated from the parallel BootGen process.
> > >
> > >
> > > I would like to clarify that I have examined both the paper [1] as well as the source code of CbAS, accessible at this link: https://github.com/brandontrabucco/design-baselines/tree/master/design_baselines/cbas, about the realm of offline model-based optimization.
> > >
> > > ---
> > >
> > > The performance disparity between BootGen and CbAS is elucidated in the results showcased in Table 1 of the main text.
> > >
> > >
> > > | Method     | RNA-A          | RNA-B          | RNA-C          | TFBind8         | GFP             | UTR             | Avg.   |
> > > |------------|----------------|----------------|----------------|-----------------|-----------------|-----------------|--------|
> > > | CbAS       | 0.541 ± 0.042  | 0.647 ± 0.057  | 0.644 ± 0.071  | 0.913 ± 0.025   | **0.865 ± 0.004** | 0.692 ± 0.008   | 0.717  |
> > > | BootGen    | **0.902 ± 0.039** | **0.931 ± 0.055** | **0.831 ± 0.044** | **0.979 ± 0.001** | **0.865 ± 0.000** | **0.865 ± 0.000** | **0.895**|
> > >
> > > Note the following table presents the experimental results on 100th percentile scores. The mean and standard deviation are reported for 8 independent solution generations. The best-scored value is marked in bold.
> > >
> > > [1] Brookes, David, Hahnbeom Park, and Jennifer Listgarten. "Conditioning by adaptive sampling for robust design." International conference on machine learning. PMLR, 2019.

---

> > > > ### Author Response · Authors · 2023-08-20
> > > > **Reminder**
> > > >
> > > > I would like to inform you that there are approximately 17 hours remaining until the conclusion of the discussion period. Could you kindly let us know if all of your concerns (about similarities with CbAS) have been addressed or if there are any remaining points of concern? Your response would greatly contribute to the progress of our paper. Thank you very much for your courteous cooperation.

---

> > > > > ### Comment · Reviewer_M7pC · 2023-08-21
> > > > >
> > > > > I read other review and rebuttal, which indeed clarifies some of my other concerns. So I increase my rating. But still, I think the CbAS and your method are very similar, but I am not that sure.

---

> > > > > > ### Author Response · Authors · 2023-08-21
> > > > > >
> > > > > > We deeply appreciate your response and support for the acceptance of this paper. To enhance its clarity, we intend to improve the "Related Works" and discussion sections. This will facilitate a more comprehensible comparison with methods like CbAS and GENhance. Your invaluable feedback has been instrumental in refining our work, and we are grateful for your continued guidance as we progress.

---

### Official Review · Reviewer_jSje · 2023-07-04

**Soundness:** 3 good
**Presentation:** 3 good
**Contribution:** 2 fair
**Rating:** 7
**Confidence:** 3

**Summary:**

This paper proposes to solve the problem of generating novel objects (in this case biological sequences) by learning weighted MLE models, and augmenting the training set of those MLE models with virtual data whose score is based on extrapolations from a proxy.

This method is tested on standard biological sequence generation problems and found to yield good scores and diversity.

**Strengths:**

I think the main strength of this paper is it confirms that extrapolation through MBO-style proxy+generative model that's recently gained popularity still can be improved and explored in a variety of ways.

The paper is well written and it was easy for me to understand the method.

**Weaknesses:**

I think the main weakness of this work is that it feels like an aggregation of methods that work well together, rather than a strong single contribution that deeply improves our understanding of MBO.
This is seen fairly well in e.g. Tables 4.3 and 4.4, there's an accumulation of things that improve performance; while it is commendable to improve the performance of ML methods, it's not clear exactly what we've learned from this paper. If there's one way I think the authors could improve this paper it's by either showing or arguing that there is a central contribution, a nugget of knowledge gained by doing these experiments.


**Questions:**

I don't have too many questions, but I have a minor concern about the fairness of comparisons; I wonder if all baselines see the same amount of data and receive roughly equivalent amounts of compute. In particular, the BootGen experiments seem to leverage an ensemble of generators ("we gather cross-aggregated samples from multiple score-conditioned generators"). Many of the baselines could also be trivially improved via ensembling; is this comparison done?

Small comment: I'd suggest using shapes (in addition of colors) in Fig 4.2


**Limitations:**

Yes, the authors address limitations.

---

> ### Author Rebuttal · Authors · 2023-08-06
>
> Thanks for providing a constructive review.
>
> **
>
> **What can we learn from this paper?**
>
> Focusing on fundamental and rigorous approaches is more important in offline design optimization tasks than relying on fancy techniques.
>
> Offline design optimization is inherently challenging as it prohibits access to the Oracle score function during training. As a result, achieving statistically stable and powerful extrapolation over the offline dataset becomes crucial. The main contribution of this study is the reinterpretation/transformation of traditional statistical concepts, such as ensemble and bootstrapping, into deep learning forms for utilization in offline design optimization. The ensemble technique can be employed as an exploration strategy to explore various modes effectively. Bootstrapping demonstrated its effectiveness as an exploitation strategy to reach high-score regions with limited samples. Particularly, through various experiments, it has been proven that the combination of these two simple yet robust concepts can serve as a powerful offline design optimization technique.
>
> 1. **Rank-based weighting**: Make a generative model focusing on the high-score regions on the offline dataset
> 2. **Score conditioned generator**: Make a conditional generative model to learn to map score to a sequence which learns the relationship between score and sequence and can extrapolate to high score region by leveraging the relationship between low-scored data of the offline dataset
> 3. **Bootstrapping using 1 and 2**: Augment training dataset using 1,2 and proxy score function to make the generative model more confident by distillation knowledge of proxy and generator itself.
> 4. **Diverse aggregation**: stabilize possible risk from 3 and leverage multiple diversified generators (trained on possible different bootstrapping scenarios) and diversified sampling.
>
> Our algorithm structure seamlessly integrates and can be viewed as a deep learning-based renovation of the classical ensemble approach involving bootstrapping and aggregation.
>
> In conclusion, our paper demonstrates the importance of carefully approaching and revisiting simple classical schemes within the context of deep learning tasks. This approach proves to be more powerful than relying on fancy techniques, as evidenced by our experiments' clear and compelling results.
>
>
> ---
>
> **Answers to the questions**
>
> Our experimental setting ensures fairness in two key aspects. Firstly, all baselines use the same offline dataset, and none of them have access to the Oracle score function, ensuring a level playing field for comparison. Secondly, the computation time is similar for every task, with almost all tasks completed within an hour (our method is notably fast).
>
> Compared with the most recent offline model-based optimization work, BDI [1], BootGen demonstrates faster speed, as shown in the table below.
>
>
>
> | | Training time|
> | -------- | -------- |
> | BDI     |  1h    |
> | Ours (1 gen)    | 3min   |
> | Ours (8 gen)    | 24min   |
>
> The speed of BootGen is measured per one generator, and we utilize 8 generators in parallel. If we were to measure the speed as serial computation (worst case), the total time taken would be approximately $3 \times 8 = 24$ minutes.
>
> It's important to note that computation time is not the main focus in offline black-box optimization tasks since the black-box score function becomes the bottleneck for computation. The primary concern lies in efficiently handling the objective evaluation process, which often requires significant time and resources.
>
> ---
> ### Response to the review of "Many of the baselines could also be trivially improved via ensembling."
>
> Our method introduces novelty through the ensemble process, which involves creating ensembles from multiple bootstrapping generators. While ensemble techniques are widely used in this field, our approach demonstrates that the direct application of ensembles does not trivially improve performance, as observed in other baselines  [2].
>
> |  | UTR (100th percentile)|
> | -------- | -------- |
> | Grad.  [2]   | 0.695 $\pm$ 0.013   |
> | Grad. (Mean ensemble)   [2]  | 0.696 $\pm$  0.009     |
> | Grad. (Min ensemble)  [2]   | 0.693 $\pm$ 0.010    |
> | BootGen (w/o DA ensemble)     | 0.729 $\pm$ 0.074   |
> | BootGen (w DA ensemble)     | 0.858 $\pm$ 0.003     |
>
> Please note that "Grad." refers to the gradient ascent method mentioned in [2]. The "Mean ensemble" represents an ensemble of proxy models that select the score value based on the mean value among proxies, while the "Min ensemble" selects the score value from the minimum value among proxies. The experimental results of the "Grad." method can be found in Table 5 of [2].
>
>
> ---
>
> Thank you for pointing out the typo; we have now corrected it in the main paper.
>
> ---
>
> ### References
>
> [1] Chen, Can, et al. "Bidirectional learning for offline infinite-width model-based optimization." Advances in Neural Information Processing Systems 35 (2022): 29454-29467.
>
> [2] Trabucco, Brandon, et al. "Design-bench: Benchmarks for data-driven offline model-based optimization." International Conference on Machine Learning. PMLR, 2022.

---

> > ### Comment · Reviewer_jSje · 2023-08-17
> >
> > Thanks for the precisions and extra data points. I think your contribution is good, perhaps ~worrying~ writing could be improved somewhat to really distill these points (but I realise this is easier said than done).
> > I will raise my score from 6 to 7.

---

> > > ### Author Response · Authors · 2023-08-18
> > >
> > > Thank you for elevating the scores, providing invaluable feedback, and supporting the acceptance of our papers.

---

### Official Review · Reviewer_ogL2 · 2023-07-05

**Soundness:** 3 good
**Presentation:** 3 good
**Contribution:** 2 fair
**Rating:** 6
**Confidence:** 4

**Summary:**

The paper proposes BootGen, a model-based sequence optimization algorithm, the author apply to the task of biological sequence design. BootGen has two stages where the first stage trains multiple sequence generators to give higher probability to sequences predicted to have higher scores from a proxy score model. The second stage uses bootstrapping by way of the trained generators to generate more data and train new generators. Samples are then aggregated in a way to encourage diversity and high fitness. BootGen is compared against previous sequence design methods on six biological sequence tasks on which BootGen achieves state-of-the-art results. Ablations show each component of the method -- bootstrapping, filtering, and rank-based reweighting -- to help performance.

**Strengths:**

- The principle behind BootGen is simple: use bootstrapping to distill signal from multiple models into one. The consensus between all the models will reduce variance and improve performance. Seeing that BootGen can outperform arguably more complicated methods is good to see.

- The combination of steps -- ranking, bootstrapping, filtering, and diversity aggregation -- are logical and synergistic. It's not a surprise they help each other.

**Weaknesses:**

- **The novelty is overstated**. Bootstrapping is a standard statistics idea, score-wise rank weights was proposed in prior work. The idea of filtering and diversity aggregation are not novel either. Adalead for instance will use the proxy model to select its final sequences. Aggregation is simply taking the union over all the samples. I don't believe there is technical novelty other than bringing together existing ideas in a straightforward way. A bioinformatics journal seems more appropriate for this work.

- There seems to be a large jump in performance by using the proxy model to filter the sequences. In fact, the performance of the method seems extremely dependent on the accuracy of the proxy model. It is not stated that the same proxy model was used across all methods in the experiments. **This leads to a unfair comparison**. I believe a fair comparison would require using the same proxy model to isolate the methodological cotributions put forth in BootGen.

- The amount of bootstrapping seems very high. The appendix states 1280 samples are needed from each generator. The number of generators is not specified clearly. The exact number of model evaluations is unclear and this would have been important to analyze. Otherwise it is difficult to tell if the method is working well due to lots of compute over other methods.

**Questions:**

- I am confused by Algorithm 2 line 2. It says N generators are trained but N is the also the number of training examples. Is this saying the number of generators trained is the same as the number of examples?

- Can the authors put the main hyperparameters in the main text? It is important to know what N, I, M are to be able to put in context how expensive the method is next to the results.

- Appendix A.3 was hard to read. "Bootstrapping is applied with 2, 500 in additional steps" what does this mean? It is important to put variables to these numbers and explain where in the method they are used.

- As stated above, was the same proxy model used across all proxy-based baselines? i.e. GFN?

- How are diverse are the generators? I would suspect they all generate similar sequences especialyl given the small dataset. Was this not a problem?

- Can the authors put some representative sequences in the appendix? It is surprising to me the method can get such high fitness in UTR with high novelty, i.e. almost half the sequence changing. Are you sure it's not adversarial examples that are being generated?

**Limitations:**

The authors mentioned the limitation of a poor proxy that could "backfire" and cause BootGen to perform worse. However, this is a general limitation for all proxy based methods. I believe there is a bigger limitation with regards to the run-time of the method with needing lots of samples and lots of training.

---

> ### Author Rebuttal · Authors · 2023-08-06
>
> Thanks for the constructive review and feedback. We provide responses for addressing the concerns below.
>
>
> ### Response for Novelty
>
> We made a novel combination of well-known techniques. Our novel high-level algorithm involves (a) propelling the generator to discover novel data points beyond the offline dataset by distilling high-likelihood high-score regions with self-confidence and (b) aggregating samples from multiple self-confident generators to minimize potential risks from adversarial samples. This algorithmic strategy consistently outperforms every baseline across all six benchmarks, highlighting the diversity and novelty of the generated samples.
>
> It would be greatly appreciated if you could note that numerous machine learning studies continue to explore innovative amalgamations of established techniques, as exemplified by the seminar paper "Rainbow: Combining Improvements in Deep Reinforcement Learning" [4]. In a similar vein, prior works like GFN-AL [1] ("Biological Sequence Design with GFlowNets") integrate the GFN [5] framework with the concept of active learning (AL) to address biological sequence design challenges. It's important to highlight that this choice could be attributed to several factors while published in ICML rather than a dedicated bioinformatics journal. Firstly, the intricacies of biological sequence design pose a significant challenge for the machine learning community. Secondly, leveraging existing ML techniques for these intricate applications offers a platform for validation and testing. Lastly, the multitude of insights and experimental outcomes from these endeavors can serve as inspiration for the wider ML community.
>
> ---
>
> ### Unfair Comparison for Proxy Usage
>
> Our proxy model follows the same structure as a recent prior work [1], a simple MLP regressor trained on the offline training dataset.
>
> ---
> ### Training time for the BootGen
>
> BootGen exhibits impressive training efficiency. Our comprehensive process capitalizes on parallel execution through eight generators, enabling the concurrent generation of 1280 samples in under a second. **Consequently, the impact of bootstrapping on training time remains minimal**. A comparative analysis against the latest offline model-based optimization work, BDI, highlights BootGen's superior speed, as demonstrated in the table below.
>
>
>
> | | Training time|
> | -------- | -------- |
> | BDI  [2]   |  1h    |
> | Ours (1 gen)    | 3min   |
> | Ours (8 gen serially)    | 24min   |
>
> This experiment is done with a single Nvidie A100 GPU. BootGen's training speed is measured per generator. When utilizing eight generators in series, the total training time amounts to approximately 24 minutes. However, this training time can be significantly reduced by adopting parallel generator training.
>
> It's worth noting that training time is not the primary focus of offline design optimization methods. In real-world scenarios, the objective evaluation, such as testing protein expressivity or binding activity, often becomes the bottleneck, taking days or even months.
>
>
>
> ---
>
> ### Answer for each question
>
>
> **A1.** Thank you for the clarification. We will avoid using the letter "N" to prevent confusion and explicitly state that we leverage 8 generators.
>
> **A2.** We appreciate the suggestion. We will include the hyperparameters as a table in the main text for better accessibility; see the attached PDF.
>
> **A3.** We acknowledge the oversight and apologize for any confusion. The bootstrapping process involves two samples with 500 iterations each, resulting in a total of 1000 bootstrapped samples. We will revise the explanation and include it in the main text.
>
> **A4.** Understood, and thank you for clarifying. We use the same proxy model as GFN-AL [1] to ensure a fair comparison. While other baselines may have slightly different learning methods for their proxy models, the core architecture remains consistent with a 2-layer MLP with a 2048 width.
>
> **A5.** You are correct. Each generator focuses on specific regions, which may lead to limited diversity compared to a highly diversified generator like GFN. However, the parallel bootstrapping process ensures that each generator concentrates on different regions, contributing to enhanced diversity. When aggregating samples from multiple generators, we can achieve better diversity than GFN-AL [1].
>
> **A6.** Thank you for the suggestion, and we look forward to the representative sequences in the appendix. Regarding adversarial examples in the context of offline design optimization, the definition involves two conditions: (1) the proxy model giving a high score, and (2) the oracle score function providing a low score. In the case of the offline model-based optimization (MBO) benchmark [3], where we assume a given oracle score function (pretrained ResNET for the 280,000 UTR dataset), our situation does not meet the criteria for adversarial examples. Both the proxy function and the oracle function provide high scores.
>
> ---
>
> ### References
>
> [1] Jain, Moksh, et al. "Biological sequence design with gflownets." International Conference on Machine Learning. PMLR, 2022.
>
> [2] Chen, Can, et al. "Bidirectional learning for offline infinite-width model-based optimization." Advances in Neural Information Processing Systems 35 (2022): 29454-29467.
>
> [3] Trabucco, Brandon, et al. "Design-bench: Benchmarks for data-driven offline model-based optimization." International Conference on Machine Learning. PMLR, 2022.
>
> [4] Hessel, Matteo, et al. "Rainbow: Combining improvements in deep reinforcement learning." Proceedings of the AAAI conference on artificial intelligence. Vol. 32. No. 1. 2018.
>
> [5] Bengio, Emmanuel, et al. "Flow network based generative models for non-iterative diverse candidate generation." Advances in Neural Information Processing Systems 34 (2021): 27381-27394.

---

> > ### Comment · Reviewer_ogL2 · 2023-08-16
> > **Response**
> >
> > Thank you for the reply. I have read the rebuttals.
> >
> > In principle, I enjoy it when simpler ideas outperform more complicated ideas and welcome it. However, the paper claims BootGen is a "novel algorithm" (line 41) then says "novel variation" line (43) and "novel bootstrapping strategy" (line 104). I don't agree it is a "novel algorithm" but agree it is a novel combination of ideas that out performs more complicated methods.
> >
> > The novelty would be more understandable if the related works was written more carefully. In particular, section 2.1 on existing protein optimization methods ends by stating, "These methods make different assumptions on the cost of evaluating the ground truth function, e.g., online optimization, sample-efficient optimization, and offline optimization." Which is not appropriate given the similarity to CbAS as reviewer M7pC stated and the similarity to AdaLead as I have stated.
> >
> > Furthermore, upon another read it is odd you only report diversity and novelty for UTR. Can the authors comment?

---

> > > ### Author Response · Authors · 2023-08-16
> > >
> > > Thank you for giving feedback on our paper.
> > >
> > > We agree with your comment that our algorithm is a novel combination; we will relax our statement regarding the novelty explanation in the main text.
> > >
> > > We will update related works more carefully, particularly comparing with CbAS, MIN, GFN-AL, AdaLead, etc., based on the discussion with you and reviewer M7pC.
> > >
> > > Regarding the diversity and novelty analyses across different tasks, we have thoughtfully included the results in Appendix C. We have conducted a thorough comparison with GFN-AL [1], which is a prominent baseline that places emphasis on diversity and novelty. This meticulous evaluation helps us provide comprehensive insights into the comparative strengths of our approach.
> > >
> > > [1] Jain, Moksh, et al. "Biological sequence design with gflownets." International Conference on Machine Learning. PMLR, 2022.

---

> > > > ### Author Response · Authors · 2023-08-16
> > > > **Methodological difference between AdaLead and BootGen**
> > > >
> > > > We extend our apologies for not previously addressing the comparison between BootGen and AdaLead [1] in our initial response. We appreciate your observant feedback that drew attention to this oversight.
> > > >
> > > > AdaLead stands as a simple yet resilient evolutionary algorithm designed for biological sequence datasets. Operating under a greedy algorithmic framework, AdaLead perturbs the most superior gathered sequence while effectively controlling the adaptive threshold denoted as $\kappa$." This approach fosters both exploratory behavior and diversity within the algorithm's operations.
> > > >
> > > > In a similar vein, BootGen operates but with a distinct approach. Unlike mutation-based exploration, BootGen's generator employs an autoregressive search across the global space rather than confining itself to local mutations. This exploration is achieved by evaluating the generator against a scoring condition, facilitating exploration by querying rather than exclusively relying on mutations of pre-collected high-scoring sequences.
> > > >
> > > > While both AdaLead and BootGen adhere to a hill-climbing philosophy, the nuances of their respective exploration methods set them apart.
> > > >
> > > > [1] Sinai, Sam, et al. "AdaLead: A simple and robust adaptive greedy search algorithm for sequence design." arXiv preprint arXiv:2010.02141 (2020).

---

> > > > > ### Comment · Reviewer_ogL2 · 2023-08-17
> > > > > **Response**
> > > > >
> > > > > Thank you. I see the comparison with GFN-AL for diversity and novelty but why did you only compare diversity and novelty with GFN-AL? GFN-AL is not the best out of the baselines as you show in table 2. A thorough evaluation would have been to compare these metrics across all the baseline. Notably Adalead is not in the results though in their paper Adalead reports getting higher performance on TF-bind and RNA than some of the baselines in this work (see table 1 of [1]). I kept my original score with the limitation being a thorough evaluation and comparison which needs to upweighted if the message of the paper is to argue for a simpler method.
> > > > >
> > > > > [1] https://arxiv.org/pdf/2010.02141.pdf

---

> > > > > > ### Author Response · Authors · 2023-08-18
> > > > > >
> > > > > > Thanks for the valuable feedback.
> > > > > >
> > > > > >
> > > > > > ## Additional experimental comparison with the AdaLead
> > > > > >
> > > > > > As you suggested, we included AdaLead [1] in six offline biological sequence tasks and compared them with BootGen.
> > > > > >
> > > > > > **Experiment setting:** We meticulously recreated AdaLead by meticulously following the source code provided in the paper, available on the FLEXS GitHub repository [1]. Our approach involved constructing a proxy model using a two-layer Multi-Layer Perceptron (MLP) with a hidden dimension of 2048, aligning with the dimensions used for BootGen and GFN-AL proxy models. To initiate our experimentation, we executed AdaLead's first round, as this mirrors the process of offline optimization and aligns with the initial round of active learning. In this context, we established the starting sequence as the one with the highest score in the offline sequence.
> > > > > >
> > > > > > ---
> > > > > >
> > > > > > **Table 1. 100th Percentile score comparison with AdaLead and BootGen. Average and standard deviation is reported with 8 independent runs**
> > > > > > | Method | RNA-A |  RNA-B |RNA-C |TFbind8 |GFP |UTR |
> > > > > > | -------- | -------- | -------- |-------- |-------- |-------- |-------- |
> > > > > > | AdaLead  | 0.691 $\pm$ 0.059     | 0.630 $\pm$ 0.062  |0.605 $\pm$ 0.055   |0.962 $\pm$ 0.024     |0.841 $\pm$ 0.014     |0.631 $\pm$ 0.010     |
> > > > > > |BootGen|0.902 $\pm$ 0.039| 0.931 $\pm$ 0.055| 0.831 $\pm$ 0.044| 0.979 $\pm$ 0.001| 0.865 $\pm$ 0.000| 0.858 $\pm$ 0.003|
> > > > > >
> > > > > > ---
> > > > > >
> > > > > > **Table 2. 50th Percentile score comparison with AdaLead and BootGen. Average and standard deviation is reported with 8 independent runs**
> > > > > > | Method| RNA-A |  RNA-B |RNA-C |TFbind8 |GFP |UTR |
> > > > > > | -------- | -------- | -------- |-------- |-------- |-------- |-------- |
> > > > > > | AdaLead     | 0.407 $\pm$ 0.018     | 0.353 $\pm$ 0.029     |0.326 $\pm$ 0.019     |0.485 $\pm$ 0.013 |  0.186 $\pm$ 0.216   |0.592 $\pm$ 0.002     |
> > > > > > | BootGen  |  0.707 $\pm$ 0.005| 0.717 $\pm$ 0.006| 0.596 $\pm$ 0.006| 0.833 $\pm$ 0.007|  0.853 $\pm$ 0.017|0.701 $\pm$ 0.004|
> > > > > >
> > > > > > ---
> > > > > >
> > > > > > As shown in the table above, BootGen outperforms AdaLead. We will include these results in Table 1 and Table 2 in the main text.
> > > > > >
> > > > > >
> > > > > > ## Reason that we compare diversity and novelty with GFN-AL
> > > > > >
> > > > > > Among all the baselines listed in both Table 1 and Table 2 of the main text, only GFN-AL [2] takes into account the aspects of diversity and novelty in their experiments. The conceptual basis for GFN-AL draws from the paper titled "Flow Network-Based Generative Models for Non-Iterative **Diverse Candidate Generation**," [3] wherein the emphasis on diversity and novelty stands as a prominent benchmark for GFN-style methodologies. Our objective lies in demonstrating a direct comparison between the outcomes of diversity and novelty with those of GFN-AL, a model distinctly crafted for the generation of varied candidates. It's worth noting that when it comes to the Pareto performance on the UTR dataset (see Figure 4.2 at the paper), GFN-AL secures the runner-up position, surpassing other baselines in performance except for BootGen.
> > > > > >
> > > > > >
> > > > > > [1] Sinai, Sam, et al. "AdaLead: A simple and robust adaptive greedy search algorithm for sequence design." arXiv preprint arXiv:2010.02141 (2020).
> > > > > >
> > > > > > [2] Jain, Moksh, et al. "Biological sequence design with gflownets." International Conference on Machine Learning. PMLR, 2022.
> > > > > >
> > > > > > [3] Bengio, Emmanuel, et al. "Flow network based generative models for non-iterative diverse candidate generation." Advances in Neural Information Processing Systems 34 (2021): 27381-27394.

---

> > > > > > > ### Author Response · Authors · 2023-08-20
> > > > > > > **Reminder**
> > > > > > >
> > > > > > > In consideration of your expressed concerns regarding the scope of empirical comparisons, we have taken the initiative to conduct additional experimental evaluations using AdaLead. We are truly grateful for your valuable insights. At this juncture, as we approach the culmination of our discussion period in the next 17 hours, we kindly request your final reflections and considerations on any remaining limitations that might warrant attention. Your concluding remarks would significantly enrich the depth of our discussion. Thank you immensely for your continued engagement and gracious contribution.

---

> > > > > > > > ### Comment · Reviewer_ogL2 · 2023-08-21
> > > > > > > > **Response**
> > > > > > > >
> > > > > > > > Thanks for your hard work! I look forward to the final version with the additional discussion and results. I have raised my score 4 to 6 with the reason for 6 (instead of 7) being there's no way to see what the final manuscript would be.

---

> > > > > > > > > ### Author Response · Authors · 2023-08-21
> > > > > > > > >
> > > > > > > > > We sincerely appreciate the invaluable feedback and ongoing discussions that have played a crucial role in improving our paper. Your insights and support have been instrumental in shaping the direction of our work. The constructive dialogue we have engaged in holds immense promise for enhancing the quality of our manuscript. Rest assured; we are fully committed to diligently incorporating the insights from our discussions into the manuscript revision process. Your contributions have fueled our determination to refine the paper and deliver an even more impactful final result.

---

### Official Review · Reviewer_LsCS · 2023-07-06

**Soundness:** 2 fair
**Presentation:** 3 good
**Contribution:** 2 fair
**Rating:** 5
**Confidence:** 3

**Summary:**

The authors propose a novel algorithm:  bootstrapped training of score conditioned generators (BOOTGEN), for the offline design of biological sequences. The key idea is to enhance the score-conditioned generator by suggesting a novel variation of the classical ensemble strategy of bootstrapping and aggregating. The method requires training multiple generators using bootstrapped datasets from training and combining them with proxy models to create a reliable and diverse sampling solution.

**Strengths:**

It is interesting the paper proposes a bootstrapping strategy to augment a training dataset with high-scoring samples that are collected from the score-conditioned generator and labeled using a proxy model.

**Weaknesses:**

Experiments:
- The choice of baselines:
   - The majority of baselines seem not biological sequence model specific (biological sequence design, e.g. https://openreview.net/forum?id=HklxbgBKvr and https://arxiv.org/abs/2006.03227, many others. ), instead the baselines are mostly general optimization methods.  Applying a general method to a new application can lead to lower accuracy. The paper can be more convincing to compare with methods target on the same application.
   - The rank based weighted score idea may be partially inspired by reference [41], which is rank-based weighting scheme for training unconditional generators   See (sec 3.1). Why not compare performance against that score design?


Idea:

-  The paper is focused on high scores samples.
Not including distribution very on median and lower score region, which might harm learning of the entire distribution. The negative samples may be helpful to learn the entire distribution, e.g. those in energy based model
-  The idea (using sequence-to-score to enhance score-to-sequence) is a relatively common idea. It will be convincing to compare with similar results.

Ablation study:
- plotting the number of generators vs accuracy can be helpful to show the efficacy of the methods.

**Questions:**

See above

---

> ### Author Rebuttal · Authors · 2023-08-06
>
> ### Comparison with Biological Sequence Model-specific Baselines
>
> Thank you for your recommendation. We have already compared it with the recent state-of-the-art method GFN-AL ("Biological Sequence Design with GFlowNets") [1] in the main text, which is an optimization method for biological sequence design. Based on your suggestion, we also made a comparison with the DyNA-PPO (https://openreview.net/forum?id=HklxbgBKvr):
>
> |  | TFBind8 (100th Percentile) | TFBind8 (50th Percentile) |GFP (100th Percentile) | GFP (50th Percentile) |
> | -------- | -------- | -------- |-------- | -------- |
> | DyNa-PPO   | 0.942 $\pm$ 0.025  |  0.562 $\pm$ 0.025   |0.790 $\pm$ 0.003  |  0.790 $\pm$ 0.005   |
> | Ours   | **0.979 $\pm$ 0.001**  |  **0.833 $\pm$ 0.007**   | **0.865 $\pm$ 0.000** | **0.853 $\pm$ 0.004**     |
>
> Note many biological sequential design methods, including the papers you suggested (**Dyna-PPO**; https://openreview.net/forum?id=HklxbgBKvr, **P3BO**; https://arxiv.org/abs/2006.03227), rely on online (model-based) optimization, where continuous score function evaluations are required for model updates. In contrast, our benchmark focuses on offline design optimization, where access to the real-time Oracle score function is limited, and only the offline dataset is available for generating high-score samples.
>
> ---
>
> ### Response for "Why not compare the performance of an unconditional generator with a score-conditioned generator?"
>
> Thank you for the insightful feedback. Here are the comparison results:
>
> |  | TFBind8 (Avg. Score) |
> | -------- | -------- |
> | RR + Unconditional Generator    | 0.505 $\pm$ 0.010  |
> | RR + Score-Conditional Generator   | **0.662 $\pm$ 0.009**  |
>
> Note the RR stands for rank-based weighting.  Our approach adapts the original rank-based weighting technique, designed for unconditional generators in latent space optimization, to work with score-conditional generators. By leveraging the score-conditional generator's ability to infer high-score regions based on query conditions, we can increment the generator's performance orthogonally. This enhancement allows for a more accurate focus on high-score regions. We have conducted thorough experiments comparing rank-based weighting and high-score conditioning, demonstrating their respective advantages and contributions to the quality of generated samples. These results offer valuable insights into the effectiveness of each approach.
>
> ---
>
> Thank you for providing additional ideas for this method.
>
> 1. **Using lower score region** Our score-conditioned generator, denoted as $p(x|y)$, is trained not only on high-score data but also on low-score data. It aims to map low-score samples to their corresponding data points. By adding weighting to high-score samples, even lower-score samples are still considered probabilistically. The study aims not only to learn an accurate distribution from diverse score-related data but also to bias the distribution towards high-sample scores for design optimization and selecting good samples. It's crucial to consider reviewers' opinions and lower-score samples throughout this process, and I wholeheartedly agree with this approach.
>
>
> 2. **Comparing with common idea** First, we already empirically compare with prior common ideas (see Table 1 and Table 2). Our method builds upon the common idea of using sequence-to-score information to enhance score-to-sequence models, as studied by MIN [3] and GFN-AL [1]. In our comparisons with these methods (shown in Tables 1 and 2), our approach outperforms them significantly. The key difference from MIN is using a proxy model for training rather than inference, which is mitigated through diverse aggregation. Compared to GFN-AL, our method generates high scores and avoids underfitting issues in the low-score region. We will further discuss these differences to provide a comprehensive analysis of the strengths and weaknesses of each approach.
>
> ---
>
> ### Reponse for the additional ablation study
>
>
> Thank you for the suggestion. We conducted an ablation study on the number of generators and their performance gain on the UTR task. The results of this study will be included in the main text, Chapter 4.4. For detailed findings, please refer to the attached PDF.
>
> ---
>
> ### References
>
> [1] Jain, Moksh, et al. "Biological sequence design with gflownets." International Conference on Machine Learning. PMLR, 2022.
>
> [2] Trabucco, Brandon, et al. "Design-bench: Benchmarks for data-driven offline model-based optimization." International Conference on Machine Learning. PMLR, 2022.
>
> [3] Kumar, Aviral, and Sergey Levine. "Model inversion networks for model-based optimization." Advances in Neural Information Processing Systems 33 (2020): 5126-5137.

---

> > ### Author Response · Authors · 2023-08-20
> > **Reminder**
> >
> > I would like to inform you that there are approximately 17 hours remaining until the conclusion of the discussion period. Could you kindly let us know if all of your concerns have been addressed or if there are any remaining points of concern? Your response would greatly contribute to the progress of our paper. Thank you very much for your courteous cooperation.

---

> > > ### Comment · Reviewer_LsCS · 2023-08-22
> > > **Thank you for the rebuttal.**
> > >
> > > I confirm that I have reviewed the author's response as well as feedback from other reviewers. I appreciate the additional experiments provided. Consequently, I have adjusted the score upwards.

---

### Author Rebuttal · Authors · 2023-08-06

Thank you to all the reviewers for providing valuable and constructive feedback on our work. We sincerely appreciate the time and effort you have dedicated to reviewing our manuscript. In response to each reviewer's comments, we have provided a detailed explanation and made necessary revisions to improve the quality of our work. Additionally, we have included a PDF file containing figures that present the results of the additional ablation study, as suggested.

We made two major responses for the two main points as overall comments:

1. **Novelty of this work.**
2. **Experimental fairness in terms of training speed and proxy model.**

---

### Novelty

Our novelty lies in the high-level algorithmic structure that combines simple yet intuitive low-level techniques for effective offline design optimization. Our high-level algorithmic structure is the reinterpretation/transformation of traditional statistical concepts, such as ensemble and bootstrapping, into deep learning forms for utilization in offline design optimization. The ensemble technique can be employed as an exploration strategy to effectively explore various modes while bootstrapping demonstrated its effectiveness as an exploitation strategy to reach high-score regions with limited samples. Particularly, through various experiments, it has been proven that the combination of these two simple yet robust concepts can serve as a powerful offline design optimization technique.

---

### Fairness of experiments

We acknowledge and appreciate the questions raised by some reviewers concerning two aspects of our approach: (a) the training time and (b) the usage of a proxy model, which they believe could potentially introduce unfairness.

However, we made fair experiments because:

1. Our approach exhibits a remarkably short training time, with each generator training process completed in approximately 3 minutes.

2. We used a simple proxy model, precisely the same as prior work [1].

Note that In offline biological sequence optimization tasks, the oracle score function is a significant bottleneck due to its computationally expensive nature; in the case of drug development, this is equivalent to the expensive testing of the drug itself, that can take days or even months. Efficiently addressing this bottleneck is therefore crucial for such offline biological design.

[1] Jain, Moksh, et al. "Biological sequence design with gflownets." International Conference on Machine Learning. PMLR, 2022.

---

### Decision · Program_Chairs · 2023-09-21

**Decision:**

Accept (poster)

**Comment:**

The paper discusses a bootstrapping technique for training of a generator for offline design of biological sequences. Reviewers raised concerns, among others about experiments, ablation studies, overstating of novelty and unfair experimentation. The rebuttal was able to address these concerns. Reviewers arrived at an unanimous accept recommendation. AC doesn't see reasons to overturn an unanimous expert consensus and recommends poster because the paper targets a smaller subset of the NeurIPS community.